# Lipidomics-Based Comparison of Molecular Compositions of Green, Yellow, and Red Bell Peppers

**DOI:** 10.3390/metabo11040241

**Published:** 2021-04-14

**Authors:** Aimee K. Sutliff, Martine Saint-Cyr, Audrey E. Hendricks, Samuel S. Chen, Katrina A. Doenges, Kevin Quinn, Jamie Westcott, Minghua Tang, Sarah J. Borengasser, Richard M. Reisdorph, Wayne W. Campbell, Nancy F. Krebs, Nichole A. Reisdorph

**Affiliations:** 1Department of Pediatrics, Section of Nutrition, School of Medicine, University of Colorado, Aurora, CO 80045, USA; Aimee.Sutliff@CUAnschutz.edu (A.K.S.); Martine.Saint-Cyr@CUAnschutz.edu (M.S.-C.); Jamie.Westcott@CUAnschutz.edu (J.W.); Minghua.Tang@CUAnschutz.edu (M.T.); Sarah.Borengasser@CUAnschutz.edu (S.J.B.); Nancy.Krebs@CUAnschutz.edu (N.F.K.); 2Skaggs School of Pharmacy and Pharmaceutical Sciences, University of Colorado, Aurora, CO 80045, USA; Katrine.Doenges@CUAnschutz.edu (K.A.D.); Kevin.Quinn@CUAnschutz.edu (K.Q.); Richard.Reisdorph@CUAnschutz.edu (R.M.R.); 3Mathematical & Statistical Sciences, University of Colorado, Denver, CO 80217, USA; Audrey.Hendricks@UCDenver.edu (A.E.H.); Samuel.S.Chen@UCDenver.edu (S.S.C.); 4Department of Nutrition Science, Purdue University, West Lafayette, IN 47907, USA; Campbeww@Purdue.edu

**Keywords:** lipidomics, bell pepper, *Capsicum annuum*, metabolomics, liquid chromatography mass spectrometry (LC/MS), β-cryptoxanthin, foodomics

## Abstract

Identifying and annotating the molecular composition of individual foods will improve scientific understanding of how foods impact human health and how much variation exists in the molecular composition of foods of the same species. The complexity of this task includes distinct varieties and variations in natural occurring pigments of foods. Lipidomics, a sub-field of metabolomics, has emerged as an effective tool to help decipher the molecular composition of foods. For this proof-of-principle research, we determined the lipidomic profiles of green, yellow and red bell peppers (*Capsicum annuum*) using liquid chromatography mass spectrometry and a novel tool for automated annotation of compounds following database searches. Among 23 samples analyzed from 6 peppers (2 green, 1 yellow, and 3 red), over 8000 lipid compounds were detected with 315 compounds (106 annotated) found in all three colors. Assessments of relationships between these compounds and pepper color, using linear mixed effects regression and false discovery rate (<0.05) statistical adjustment, revealed 11 compounds differing by color. The compound most strongly associated with color was the carotenoid, β-cryptoxanthin (*p*-value = 7.4 × 10^−5^; FDR adjusted *p*-value = 0.0080). These results support lipidomics as a viable analytical technique to identify molecular compounds that can be used for unique characterization of foods.

## 1. Introduction

Nutritional research has established that certain foods and food compounds influence human health and disease; however, the biochemical underpinnings of the reported health effects require further investigation. Additionally, there is a myriad of uncharacterized food-related compounds, which together results in the delivery of incomplete, ambiguous, and confusing nutrition messages to the general public. The outcomes of food science research is an important element that contributes to the likelihood of individuals choosing to consume appropriate health-promoting foods and adhere to a healthy diet [1]. 

Distinctive characteristics of a food, such as its phenotype (e.g., color or pigmentation) confers unique health benefits, but also contributes to the complexity of the food’s molecular composition. The study of biologically active compounds, as they attribute to the sub-variety of a food, e.g., red vs. green bell pepper, is not entirely comprehensive, therefore, perpetuates the gap in our understanding of their relative health benefits. Metabolomics, which is the global study of small molecules (i.e., compounds), can help close this knowledge gap by deciphering how the molecular composition of a food differs or changes based on phenotypic characteristics such as pigmentation [2]. In addition, metabolomics may help distinguish between non-phenotypic factors such as how the food is grown, processed, shipped, and consumed [3,4,5,6,7,8]. Finally, metabolomics can be used to discover biomarkers of specific foods, food groups, or eating patterns and to identify associations with health outcomes [9]. Thus, our understanding of whether the health effects of foods are altered by the variability in their molecular composition can be enhanced by the successful application of metabolomics to provide an extensive profile of the molecular composition of a food.

The bell pepper (*Capsicum annum*) is one of the most popular consumed fruits worldwide due to its high content of bioactive compounds, strong antioxidant capacity, flavor, nutritional value, and distinctive colors [10]. Though all bell peppers, despite variations in color, are considered the same fruit, their color may confer significant variability in taste, cost, nutrient content, bioactive compounds, and antioxidant capacity [10,11,12]. The variability in pigmentation seen in bell peppers is largely due to lipid-based antioxidant phytochemicals and carotenoids rather than more water-soluble flavonoids such as anthocyanins [10,13]. Green bell peppers contain chlorophylls, which contribute to their green color along with distinctive carotenoids such as lutein, violaxanthin, and neoxanthin [14,15]. Similarly, red peppers contain capsanthin and capsorubin, whereas violaxanthin, β-carotene, lutein, antheraxanthin, and zeaxanthin are the carotenoids most common in yellow peppers [15]. The differences in the bitter versus sweet flavors of green and red peppers are due to differences in organic acids and sugars [16]. Additionally, polyphenols such as the flavonoid, quercetin, has been reported to differ in concentration in bell peppers, indicating higher concentrations in yellow and red peppers compared to green peppers [17]. While much is already known about these specific molecular components of bell peppers, an untargeted metabolomics approach could potentially yield valuable information about other, less well-characterized compounds.

The exploratory research presented here sought to determine the feasibility of using untargeted lipidomics to identify and evaluate differences in the molecular composition of green, yellow, and red bell peppers [11]. The application of an untargeted lipidomics approach to characterize a large number of compounds from a variety of bell peppers is a novelty of this research. While metabolomics often focuses on more water-soluble compounds, we aimed to determine if there were differences in color-associated compounds in bell peppers, which are more lipid-soluble, such as the carotenoids listed above. Therefore, this research applied untargeted lipodomics approach to assess the molecular composition of a lipid-rich fraction of bell peppers as they relate to color. The feasibility of this approach can be assessed, in part, by comparing previous reports on carotenoids to the current study. In the longer term, studies such as these will accelerate discovery of how food-specific compounds impact health outcomes through potential interactions with nutrients and physiological processes.

## 2. Results

### 2.1. Metabolomics Reveals Several Lipid Compounds in Pepper

A total of 23 pepper samples corresponding to six peppers were analyzed using an untargeted lipidomics approach. Following extraction of raw liquid chromatography mass spectrometry (LC/MS) data, 11,127 compounds were detected in at least 1 pepper sample and 8174 compounds detected in at least 2 pepper samples of the same color. Following data processing, a total of 315 compounds were detected in 100% of the 23 bell pepper pieces from the six bell peppers and were further analyzed. Of the 315 compounds, 106 were annotated after searching multiple metabolomics databases at a minimum of Metabolomics Standards Initiative (MSI) level 3 (low-medium confidence) [18]. The software, MetabAnnotate, was used to add descriptive information (e.g., chemical class) to any compound with an Human Metabolome Database (HMDB) 4.0 ID [19] (Appendix A, Column AA). MetabAnnotate also added compound IDs from other databases, such as FoodDB [20], PubChem [21], and KNApSAcK [22].

Annotated compounds included 57 glycerolipids and glycerophospholipids, such as triacylglycerols (TGs), diacylglycerols (DGs), phosphatidylcholines (PCs), phosphatidylinositols (PIs), phosphatidylglycerols (PGs), phosphatidic acid species (PAs), and phosphatidylethanolamines (PEs) (Appendix A). Two sphingomyelin (SM) compounds and a ceramide were also detected. Additional detected compounds included several prenol lipids, steroidal glycosides, and annonaceous acetogenins, which are waxy derivatives of fatty acids (Appendix A).

### 2.2. Principal Component Analysis (PCA) and Hierarchical Clustering (HC) Illustrate Pepper Samples Group Together Based on Color

PCA of the 8174 compounds detected in at least two of the 23 pepper samples from six peppers revealed that peppers cluster by color, with red and yellow peppers grouping more closely compared to either green and red or green and yellow peppers (Figure 1A). HC of compounds from peppers of different colors revealed regions of distinct variation, as well as similarities (Figure 1B). For instance, the region to the far left of the HC graphic reveals compounds that are highly abundant in all three pepper colors (solid black box). Variation in compound abundance between different colored peppers is also demonstrated (dashed black box). Interestingly, the compounds within the dashed box were also found to be driving some of the separation in PC1 of the green peppers from yellow and red peppers in the PCA when the PCA loadings plot was examined (data not shown).

### 2.3. Untargeted Lipidomics Reveals Several Compounds That Differ between Green, Red, and Yellow Peppers

Briefly, 8174 compounds were detected in at least 2 pepper samples of the same color (often from the same individual pepper), with 1813 compounds unique to red peppers compared to 1229 and 324 unique to green and yellow peppers, respectively. This suggests that pepper colors may have distinct chemistry (Figure 1C). While the six peppers differed in characteristics such as color, organic status, harvest location, and time of year, due to sample size and study design limitations, statistical analysis could only be used to evaluate differences due to color. A total of 111 compounds were found to be nominally associated with pepper color (*p*-value < 0.05) (Appendix A). These included a total 39 annotated compounds, which include 26 in the class of glycerolipids and glycerophospholipids and 13 miscellaneous classes (Table 1). A total of 11 annotated compounds were found to differ by color with false discovery rate (FDR) adjusted *p* < 0.05 [23] (Table 1, above bold line). This included one monoglycerolipid, two diglycerides, two triglycerides, five phospholipids, and one carotenoid (Table 1). The compound most strongly associated with color was β-cryptoxanthin (*p*-value = 7.4 × 10^−5^; FDR adjusted *p*-value = 0.0080). Other annotated compounds, such as sucrose acetate isobutyrate (SAIB), 2-ethenyl-2,4b,8,8-tetramethyl-tetradecahydrophenanthrene-3,5,10a-triol, fargesin, and ascorbyl linoleate/L-ascorbyl linoleate (LAA) were found to nominally differ by color but not FDR adjusted (Table 1, below bold line).

### 2.4. Pairwise Comparisons of Green, Yellow, and Red Peppers

Based on these results from the untargeted analysis, the 11 compounds passing an FDR of 0.05 were further evaluated using pairwise comparisons (i.e., *t*-tests). Because individual compounds were being evaluated rather than an entire dataset, non-phenotypic factors such as organic vs. non-organic and location were again considered in the analyses in addition to color. For the 11 compounds significant after FDR adjustment, the highest number of differences occurred when green and red peppers were compared (*n* = 9 compounds, Table 1, gray cells above bold line), following by green vs. yellow (*n* = 4, Table 1, gray cells above bold line) (Table 1 and expanded version in Appendix A). Only β-cryptoxanthin differed significantly between yellow and red peppers. Subsequent pairwise comparisons showed that β-cryptoxanthin levels were 12.1-fold higher in red peppers compared to green peppers and 8.2-fold higher than yellow peppers (*p* = 7.42 × 10^−5^, Figure 2A).

In addition to several compounds found to be significantly different between peppers following FDR adjustment, a number of compounds reached significance but did not pass FDR. For example, the tentatively annotated SAIB compound was nominally different between green and red peppers (*p*-value = 0.0065 and FDR = 0.056). The putatively identified compound ivermectin B1b was found to be nominally significantly different between green and yellow peppers (*p*-value = 0.0488), while all-trans-retinyl oleate was close but not below the nominally significant threshold for red vs. yellow peppers (*p*-value = 0.0523).

### 2.5. Evaluation of Pepper Attributes for Beta-Cryptoxanthin

Further analysis of the top result of β-cryptoxanthin was done since it is a “true positive” or compound known to be associated with bell peppers. The other 10 from the top 11 are ambiguous compounds, not specifically related to food and hence additional analyses was not conducted in order to avoid overinterpretation of the data. The additional analysis showed that there was no statistically significant difference in the β-cryptoxanthin levels between yellow and red cooked and raw peppers (*p*-value = 0.794) (Figure 2B) or by green and red peppers grown organically compared to conventionally (*p*-value = 0.978). Upon further inspection of the red and green peppers separately, the non-organic red pepper from Canada (three replicates) had significantly lower β-cryptoxanthin levels compared to the two organic red peppers from Mexico (three and five replicates) (*p*-value = 0.014) (Figure 2C). The non-organic green pepper from Mexico (three replicates) did not have significantly different (*p*-value = 0.544) levels of β-cryptoxanthin compared to the organic pepper from the United States (four replicates) (Figure 2D). Since these results are based on a small sample of peppers, it is not possible to discern if the differences seen in red organic status are due to differences in harvest location instead of organic vs. non-organic. There continued to be no significant difference by cooked vs. raw when assessing within yellow and red peppers (*p*-value = 0.913 and 0.660, respectively). Given the extremely small sample sizes, all of these results should be interpreted with caution and warrant further study.

## 3. Discussion

In the current study, we assessed six bell peppers with multiple characteristics such as color, organic status, harvest location, and purchase time of year. Our methods detected over 8000 compounds in peppers, with 315 compounds found in 100% of the 23 samples from the six individual peppers analyzed. The application of an untargeted lipidomics approach to characterize a large number of compounds from a variety of bell peppers is a novelty of this research. However, due to limited sample size, color was the only characteristic that could be statistically analyzed with confidence, revealing 11 compounds that reached significance and passed FDR. Interestingly, the top 11 revealed one compound, β-cryptoxanthin, as a “true positive” or previously known compound to be associated with bell peppers. The remainder of the compounds in the top 11 are commonly detected and/or ubiquitous.

Many of the compounds detected in the current study have yet to be characterized as being specifically related to bell peppers and a myriad are not present in any existing metabolomics database at the time of this study. For example, of the 315 compounds detected in all samples, only 106 were annotated from databases accessed in August 2020. While 111 compounds were found to be nominally statistically different across pepper color, only 39 were annotated and taken forward for further analysis (Appendix A). The further analysis indicated only 11 annotated compounds, including β-cryptoxanthin, which are identified from the databases used (see Appendix A for more details). Further analysis was performed on β-cryptoxanthin because it is known to be found in bell peppers and hence biological interpretation would be more reliable. β-cryptoxanthin is the only named compound that reached *p*-value significance and passed FDR; while the other named compounds have less specificity to bell peppers, our analysis nonetheless showed that they reached *p*-value significance.

Subsequent to the global statistical approach a more focused, pairwise analysis was conducted on β-cryptoxanthin. Red peppers had significantly more β-cryptoxanthin, compared to green or yellow peppers, which supports previous studies demonstrating concentration differences of this compound as a result of fruit maturity or ripening that results in bell pepper color changes [11,16]. The finding of β-cryptoxanthin as a significant difference between the samples represents the feasibility of using an untargeted approach to identify candidate molecules that can then be interrogated more deeply. Caution is required when conducting focused analyses on candidates discovered using a small sample size; in the current case, β-cryptoxanthin was previously known to be related to pepper characteristics and hence represented a “true positive,” thereby lending confidence to the results. While a number of compounds have been identified as differing across pepper colors [15,16], to our knowledge, an untargeted lipidomics strategy that focuses on lipid-based or pigment-related molecules, such as carotenoids, has not yet been reported. Our metabolomics-based approach is a “hypothesis generating” strategy that can be performed using similar techniques on other foods to identify compounds of interest.

β-cryptoxanthin is a pigment compound in the carotenoid family responsible for the orange/red color to many fruits and vegetables [24,25]. Carotenoids are categorized by the presence or absence of oxygen on the terminal ends of the molecule and are well known for their antioxidant function with variable antioxidant capacity [24]. Analytical data on the carotenoids of pepper are divergent and there are difficulties in separating, identifying conclusively, and quantifying the major carotenoids. The carotenoid action in human health is intimately related to their structure and the effects differing with different carotenoids; thus, conclusive identification and accurate quantification of individual carotenoids are necessary. For example, carotene is the primary carotenoid that contains no oxygen with several isomers, such as β-carotene, α-carotene, and γ-carotene [24]. β-Cryptoxanthin is an oxygenated carotenoid termed xanthophyll with a chemical structure similar to, but more polar or hydrophilic than β-carotene [25]. Though β-carotene is present in more fruits and vegetables than β-cryptoxanthin, β-cryptoxanthin is found at high concentrations in certain foods, has greater bioavailability, and has nearly equivalent antioxidant capacity when compared to β-carotene [25]. β-cryptoxanthin has been demonstrated to have anti-proliferative effects in lung cancer, may have hepato-protective effects, and may improve insulin resistance [26,27,28].

Previous studies have investigated different cooking methods and their impact on β-cryptoxanthin levels in foods, which have included various microwave cooking times and various boiling times [29], as well as in pepper samples that were frozen and boiled compared to raw [30]. Cooking either by boiling or microwave heating was reported to result in an approximately 20% loss in total carotenoid content [31]. Total carotenoid content of red bell pepper has been reported to be significantly reduced by boiling, steaming, and roasting, but not stir-frying, as compared to raw [32]. The current study used a very brief cook time to avoid burning. Oils were not used in the cooking process to avoid contributions of oils to the compound analysis and limit our cook time. While these previous studies demonstrated significant losses of β-cryptoxanthin as cook time was increased [29,30], our study did not reveal significant differences in either the yellow or red pepper β-cryptoxanthin levels compared to the matched raw pepper. Our data suggest that peppers may be capable of withstanding short cook times without significant losses of β-cryptoxanthin, although more research is needed with a larger sample size to confirm these findings.

Differences in carotenoid content have been reported as related to growing conditions, such as higher levels of β-cryptoxanthin in peppers grown in a glasshouse compared to those grown outdoors [33]. Organic compared to conventionally grown counterparts have been evaluated in a variety of plant foods, including bell peppers, showing that fresh vegetables from organic production are usually richer in biologically active compounds when compared to the conventional ones [34]. A study determined that pickled red bell peppers revealed significantly higher content of total carotenoids, and specifically β-cryptoxanthin, in organically grown peppers [34]. The same laboratory compared organic vs. conventional white cabbage and yet determined higher carotenoid content in the conventionally grown samples compared to organic [35]. In the current study, pairwise analysis of β-cryptoxanthin in red and green peppers showed the non-organic red pepper from Canada had significantly lower β-cryptoxanthin levels compared to organic red peppers from Mexico; however, it was not possible to definitively determine organic status as a single factor that contributed to differences in the carotenoid content between those peppers. Since our results are based on a small sample of peppers, more samples with less variation in season, location, and soil conditions would be needed to further evaluate the organic status for comparison between the peppers.

As described in the Appendix A, MS2 level annotations were obtained for only a small number of compounds (Appendix A); therefore, with the exception of α-cryptoxanthin, annotations should be considered putative with a MSI level 2 or 3 [18]. Identification of several compounds, such as 2-ethenyl-2,4b,8,8-tetramethyl-tetradecahydrophenanthrene-3,5,10a-triol, all-trans-retinyl oleate, and archaetidylglycerol-myo-inositol, lack characterization as they relate to any food, including bell peppers. Of the remaining compounds that were significantly different by pepper color (*p*-value < 0.05), several have well-described associations with foods and health benefits, such as ascorbyl linoleate/L-ascorbyl linoleate (LAA), the most biologically active and well-studied form of vitamin C [36]. Vitamin C content is higher in bell peppers compared to several other vegetables and fruits commonly recognized as vitamin C sources [10]. Vitamin C is an important antioxidant compound that supports collagen production, prevents common degenerative conditions like cataracts, and aids immune system functioning [11]. Another compound, identified as fargesin, is a bioactive neolignan isolated from magnolia plants, with antihypertensive and anti-inflammatory effects; however, it has not been previously identified in bell peppers [37,38,39].

Another compound was tentatively annotated as glycidyl oleate, which is a carboxylic ester and an epoxide. This compound has been previously described as a possible carcinogen, which may be the result of heat, food processing, or a contamination [40]. In the present study, bell peppers were minimally cooked to avoid molecular changes as a result of heat, though there have been no studies investigating glycidyl oleate composition as it relates to bell peppers and cooking. This study identified Goyaglycoside g as a compound in bell peppers; however, it has only been previously isolated from the fresh fruit of Japanese *Momordica charantia L*. (Cucurbitaceae), which has been used in several Asian cultures as a stomachic, laxative, or an anthelmintic [41]. There are known nutraceuticals and functional food peptides known as angiotensin converting enzyme inhibitors (ACE) inhibitors that belong to a group called “bioactive substances” [42]. Various plants are natural sources of ACE inhibitors such as soybean, mung bean, sunflower, rice, corn, wheat, buckwheat, broccoli, mushroom, garlic, spinach, and grapes [43]. Bell peppers may potentially be a source of ACE inhibitors as well, though this has not yet been previously described in the literature.

Additionally, our study revealed some compounds that may be potentially related to undescribed agricultural practices. For example, ivermectin B1b is a compound with antiparasitic activity that is the minor component (<20%) of the anthelmintic ivermectin, which is mainly composed of ivermectin B1a (>80%) [44]. Navratilova et al. have demonstrated that the utilization of ivermectin to prevent and treat animal parasitic diseases in livestock has the potential to enter soybean roots and leaves, but not the beans, through the use of manure for fertilizer [45]. The compound SAIB was identified in bell peppers, however, it is most commonly known for its use as an alternative food emulsifier in non-alcoholic carbonated and non-carbonated beverages [46]. Though compounds such as ivermectin and SAIB have not been previously shown to be associated with bell peppers, the findings in this study support gaps in our understanding of agriculture and food preservation practices.

This study has some limitations, such as the inherent limitations of lipidomics in terms of the resulting purity and integrity of the samples. As described in Appendix A, we were unable to distinguish between the carotenoids β-cryptoxanthin and α-cryptoxanthin using authentic standards analyzed by tandem MS or ion mobility MS. While these isomers can be resolved using different chromatography methods [47], repeating the study under different conditions was not feasible, however, α-cryptoxanthin is rarely found in plants [48]. Moreover, our stringent methods may have filtered out other carotenoids that were not abundant or present in all samples, therefore, not included in the final 315 compounds utilized in the analysis. The lipidomics approaches isolated a large number of compounds, however, most could not be annotated due to the narrow coverage of plant compounds in many metabolomics databases, therefore limiting the number of compounds available for a comprehensive analysis. Furthermore, while lipidomics is appropriate for understanding how lipid species vary between samples, it does not capture differences in non-lipid species, consequently, abating the contributions of aqueous compounds. Due to study design and sample size limitations, additional analyses with complementary and comparative methods will be needed to assess the robustness of these results as well as to accommodate for more in-depth analysis of various food characteristics, such as location and harvest time.

The study supports previous studies that have shown that red peppers had significantly more β-cryptoxanthin, compared to green or yellow pepper. Additional studies are required to further test the relationship of the compounds as they relate to bell pepper color and other variable qualities such as cultivation, harvest, processing, storage, and cooking. Lipidomics can help close this knowledge gap by deciphering if the molecular composition of a food is altered based on these and other factors. The compounds related to the distinctive characteristics of an individual food may suggest unique health benefits. Thus, determining whether the health impact of foods is altered by the variability in their characteristic may be a valuable target for future nutritional research.

Overall, this research serves as a proof-of-principle that further characterization of bell peppers can be determined by applying untargeted metabolomics to identify compounds associated with specific food qualities, such as pigmentation. This research also demonstrates the utility of starting with an untargeted approach to identify potentially interesting molecules, followed by a targeted approach to look more specifically at single compounds of interest, such as β-cryptoxanthin. Hence, our metabolomics-based “hypothesis generating” strategy can be performed using similar techniques on other foods to identify compounds of interest.

## 4. Materials and Methods

### 4.1. Chemicals, Standards, and Reagents

All solvents used for small molecule (compound) extraction and LC/MS analysis were of high-performance liquid chromatography (HPLC) or LC/MS-grade, which included: water and isopropyl alcohol from Honeywell Burdick & Jackson (Muskegon, MI, USA) and methanol, chloroform, formic acid, acetonitrile, and methyl tert-butyl ether (MTBE) from VWR (Radnor, PA, USA). Authentic standards for sample preparation were from Avanti Polar Lipids Inc. (Alabaster, AL, USA), Cambridge Isotope Laboratories (Tewksbury, MA, USA), and Sigma Aldrich (St. Louis, MO, USA).

### 4.2. Bell Pepper Sample Preparation

Green, yellow, and red bell peppers were purchased locally in Denver, Colorado, U.S.A., and represented different farming practices, locations, and purchase time (Table 2, *n* = 1 each with technical replicates). Green and red peppers that were conventionally grown in Mexico and an organic variety grown in the United Stated, organic red peppers grown in Mexico, and conventionally grown peppers from Canada were purchased from the same grocery store. Four months later, one organic, Mexico grown red pepper and one yellow, organic pepper, with no label indicating company name or country of origin, were purchased together from a different location than the peppers purchased previously.

All peppers were chopped into similar sized pieces and an equal aliquot was stored in 15 mL Eppendorf tubes. The red and yellow peppers purchased together in March 2019 were each split into two groups: (1) raw and (2) cooked separately over medium heat in a multicooker with no oil or additives for 42 s while continually stirred to avoid charring. All samples were stored at −80 °C until lyophilization. All pepper samples were lyophilized for 72 h at −40 °C. Dried bell pepper samples were suspended in ice cold 100% methanol at 100 mg/mL and bead homogenized for two cycles of 5 min at 50 Hz on a Qiagen TissueLyser LT (Germantown, MD, USA).

For this untargeted lipidomics approach, a modified liquid–liquid extraction method using MTBE was used to separate hydrophobic and hydrophilic fractions of each sample [49,50]. Briefly, 100 µL pepper homogenates (10 mg lyophilized equivalent) were spiked with 10 µL Avanti Polar Lipid’s SPLASH Lipidomix prior to extraction. A three-to-one mixture of MTBE-to-methanol was added to each tube and then vortexed. Then, a three-to-one mixture of water-to-methanol was added and vortexed. The samples were centrifuged at 4 °C for 15 min, followed by removal of the lipid and the aqueous layer separately and dried under nitrogen. Lipids were reconstituted in 100 µL of 100% methanol; aqueous fractions were reconstituted in 50 µL of 95% water and 5% acetonitrile, as previously described [49]. Only the lipid fractions were used for the purposes of the current study. The hydrophilic fraction was stored at −80 °C for future analysis and publication.

### 4.3. Liquid Chromatography (LC) Mass Spectrometry (MS)

Samples were analyzed using an Agilent 6545 liquid chromatography-quadrupole time-of-flight mass spectrometry (LC-QTOF-MS) (Agilent Technologies, Santa Clara, CA, USA). Lipid fractions of all pepper samples were analyzed using reverse-phase chromatography with an Agilent Zorbax Rapid Resolution HD SB-C18, 1.8 µL m (2.1 mm × 100 mm) analytical column. Injection volume was 4.0 µL with a flow rate of 0.7 mL/min. Mobile phase A consisted of water with 0.1% formic acid, while mobile phase B consisted of 60:36:4 isopropanol:acetonitrile:water with 0.1% formic acid. The gradient was as follows: 30% B to 70% B over 0.5 min, increase to 100% B over 7.4 min, hold at 100% B for 5 min, and decrease back to 30% B over 0.1 min. The total run time was 17.1 min with 4.5 min postequilibration time. The autosampler tray temperature was set to 4 °C with a column temperature of 60 °C.

### 4.4. Mass Spectrometry (MS)

Samples were analyzed using an Agilent 6545 Time-of-Flight (TOF-MS) with dual Agilent JetStream electrospray ionization (AJS-ESI) source in positive ionization mode. Specific parameters are as follows: scan rate of 2 spectra/s, mass range of 75–1700 *m/z*, drying gas temperature 300 °C and flow rate of 12.0 L/min, nebulizer pressure 35 psi, sheath gas temperature 275 °C, sheath gas flow 12 L/minute, skimmer 65 V, capillary voltage 3600 V, fragmentor 100 V, and reference masses 121.050873 and 922.009798 (Reference mix, Agilent Technologies, Santa Clara, CA, USA).

### 4.5. Data Processing

Raw LC/MS data were extracted using Agilent Technologies MassHunter Profinder Version B.08 (Profinder) software and analyzed using Agilent Technologies Mass Profiler Professional Version 14.1 (MPP), as previously described [9]. Untargeted and recursive feature extraction was applied to compound data from each sample, using abundance profiles in m/z and retention time (RT) dimensions. Lipid positive mode samples were extracted using Batch Molecular Feature Extraction (BMFE) in Profinder with the following parameters: retention time extraction range of 0–10.4 min with noise peak height filter ≥ 3000 counts; ion species: +H, +Na, +K, +NH_4_; and charge state maximum of two. Alignment tolerance for RT was 0% + 0.3 min with mass 20 ppm + two mDa (millidalton). BMFE parameters were height at ≥ 15 counts and a score of ≥ 50. Compounds were then imported into MPP and filtered based on presence in at least two samples. Any compounds detected in the spiked blank samples were removed. The remaining list of compounds was then imported into Profinder for searching with Batch Targeted Feature Extraction (BTFE) using the following parameters: height ≥ 10,000 for EIC peak integration with post-processing filters using Abs height ≥ 10,000 counts and score ≥ 50.

### 4.6. Compound Annotation

Processed data were reimported back into MPP for annotation using Agilent MassHunter ID Browser B.08 (ID Browser) to search in-house and commercial databases. The in-house database is composed of HMDB 4.0 [19], Lipid Maps [51], National Institute of Science and Technology (NIST) [52], and a set of 683 authentic standards with tandem MS analysis (MS/MS) data. Annotations were based on accurate mass, with a mass error cutoff of 10 ppm, isotope ratios and isotopic distribution through which the predicted isotope distribution is compared to actual ion height and a score is generated. Scores > 50 were considered putative annotations and correspond to an MSI metabolite identification level two or three [18]. In addition, unmatched data for significant compounds were manually searched using FoodDB [20], Phenol-Explorer [53], and KNApSAcK [22]. For compounds in which no annotation was possible, the molecular formula generator in ID Browser was used to estimate a metabolite chemical formula. All data and annotations were also manually reviewed.

To aid in interpretation of data, a software tool was developed to efficiently annotate identified compounds. The tool, MetabAnnotate, along with open-source Python 3 code is available at https://github.com/AnachronicNomad/metabolite_annotations (accessed on 2 February 2021). Briefly, MetabAnnotate takes an input Excel file, uses batch export by HMP ID to access XML files from the HMDB [19]. Project database v4 is found at https://hmdb.ca/metabolites, and it outputs an annotated Excel file that includes the HMDB [19] description as well as other database IDs such as Kyoto Encyclopedia of Genes and Genomes (KEGG) [54] and Lipid Maps [51].

### 4.7. Tandem MS (MS/MS)

To improve confidence in annotations, MS/MS was performed by targeting m/z and retention time (RT) of statistically significant compounds of interest. Lipid extracts were analyzed using the above LC-MS method with MS/MS data collected at fixed 10, 20, and 40 eV collision energies. Resulting experimental MS/MS spectra were compared to the National Institute of Science and Technology (NIST) Tandem Mass Spectral Library (Version 2.3) [52,55] using the NIST14 and NIST17 MSMS spectral libraries. The data were then searched using the in silico MS/MS spectral interpretation software SIRIUS version 4.6.0 [56] for formula and CSI:FingerID version 1.4.8 [57] for compound annotation. To increase confidence in compound ID database searches, pepper compounds were limited to the following natural product databases: Collection of Open Natural Products (COCONUT) [58], Global Natural Products Social Molecular Networking (GNPS) [59], Plant Metabolic Network (PMN) [60], KNApSAcK [22], and SUPER NATURAL II [61]. In silico strategies for improving confidence in annotations are further described in the Appendix A. For example, one molecule of interest was initially annotated as β-cryptoxanthin although manual review of the data showed that α-cryptoxanthin is indistinguishable from β-cryptoxanthin, both of which are 552 *m/z*, though the latter is rarely found in plants [48]. As described in the Appendix A, additional mass spectrometry, in silico strategies, and manual examination of the data resulted in confirmation of the original annotation of β-cryptoxanthin.

### 4.8. Statistical Analysis

Statistical analysis was performed in R Studio using R v.3.5.1. A linear mixed effects model using the function *lmer* from the lme4 [62] package was used with log2 transformed relative metabolite peak height (i.e., comparing relative abundances) as the outcome and pepper color as the predictor. A random intercept term for pepper was used to control for correlation due to pepper pieces being from the same individual pepper. An adjusted model controlling for organic status and raw vs. cooked was used to assess stability of the results. A FDR of 0.05 [23] was used to identify compounds significantly associated with pepper color. Post-hoc pairwise comparisons for pepper color were performed with Tukey’s multiple testing correction using the *emmeans* function from the emmeans package. For the compound most significantly associated with pepper color, additional analyses were performed on overall pepper colors and within pepper color where it was possible to evaluate other pepper attributes such as organic status and cooked vs. raw. Similar to before, linear mixed effects models with a random intercept term for pepper were used.

### 4.9. Data Visualizations

Visualization of the data using principal component analysis (PCA) and hierarchical clustering (HC) were performed in MPP [9]. For PCA, log2 transformed data were mean centered scaled and the PCA was performed using the non-averaged sample groups interpretation with pruning using 4 principal components. For HC, data from 8174 compounds found in at least two samples were clustered on both samples and compounds and performed using averages of replicates within each color. The MPP software for HC uses using log2 transformed intensity values with the Euclidean distance matrix algorithm and Ward’s linkage rule.

## Figures and Tables

**Figure 1 metabolites-11-00241-f001:**
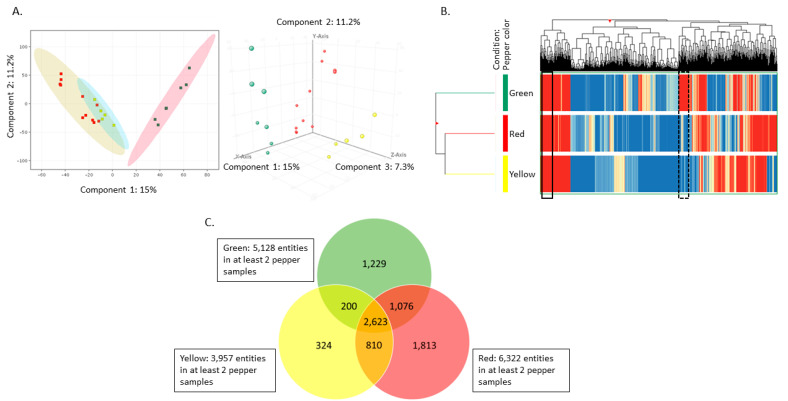
Pepper color is the main driver of differences between pepper conditions. (**A**) Principal component analysis (PCA) was performed in Agilent Technologies Mass Profiler Professional Version 14.1 (MPP) using data from all bell pepper samples. Component 1, which explains 15% of the variation, is shown on the *x*-axis; component 2, which explains 11.2% of the variation, is shown on the *y*-axis; and component 3, which explains 7.3% of the variation, is shown on the *z*-axis. (**B**) Hierarchical clustering of data from 23 replicates from six individual bell peppers. The *x*-axis corresponds to individual compounds detected in the peppers, which are grouped by color and listed on the *y*-axis. Blue lines indicate less relative abundance of a compound compared to the other 8174 compounds, while red lines indicate higher relative abundance compared to the other 8174 compounds. The vertical distance between compounds provides a rough estimation of their similarity. (**C**) Venn diagram illustrates overlap between the 2623 compounds detected in all colors of pepper samples (center section), the 1229 compounds detected in green peppers (green circle), but not red or yellow; the 200 compounds in both green and yellow; and the 1076 compounds in both green and red pepper. Within the yellow peppers (yellow circle), 324 compounds were detected in yellow but not red or green and 810 compounds were detected in both yellow and red bell peppers. Red peppers contained 1813 compounds that were detected in red (red circle), but not green or yellow bell peppers.

**Figure 2 metabolites-11-00241-f002:**
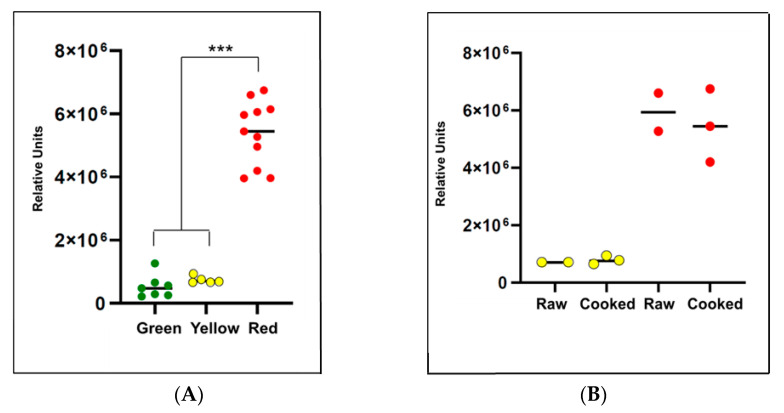
Relative abundance of β-cryptoxanthin detected in bell pepper samples by color. Following analysis of pepper samples using untargeted LC/MS, β-cryptoxanthin levels were compared according to their relative abundance across samples. (**A**) Red bell pepper has significantly higher levels compared to green and yellow. *** *p* < 0.001; (**B**) To determine whether cooking had any impact, each pepper was divided, three samples were heated for 5 min, and two samples remained raw. There was no statistically significant difference between the cooked peppers compared to raw overall or within either pepper color; (**C**) There was a nominal statistical difference between one non-organic, Canadian red pepper compared to two organic, Mexican red peppers; (**D**) There was no statistical difference between the non-organic, Mexico grown green pepper vs. the organic US grown green pepper.

**Table 1 metabolites-11-00241-t001:** Annotated pepper compounds nominally associated with color.

Compound Name	*p*-Value *	FDR	Green vs. Red ^	Green vs. Yellow ^	Red vs. Yellow ^	Class
Beta-Cryptoxanthin	0.000074	0.0080	0.010	0.677	0.035	Carotenoid
PC(O-20:0/O-1:0)	0.000128	0.0080	0.012	0.459	0.059	Phospholipid/PC
PC(O-20:0/O-1:0)	0.000129	0.0080	0.012	0.454	0.059	Phospholipid/PC
PI(18:2/13:0)	0.000154	0.0080	0.021	0.035	0.384	Phospholipid/PI
DG(22:6/18:0)	0.000769	0.0186	0.705	0.047	0.060	Glycerolipid/DG
PG(22:2/20:0)	0.001613	0.0318	0.052	0.117	0.999	Phospholipid/PG
DG(20:5/14:0)	0.002207	0.0366	0.053	0.071	0.761	Glycerolipid/DG
TG(22:2/15:0/20:5)	0.003347	0.0458	0.001	0.077	0.390	Glycerolipid/TG
PI(P-20:0/14:1)	0.003822	0.0458	0.014	0.002	0.331	Phospholipid/PI
MGDG(18:1/18:1)	0.003996	0.0458	0.999	0.095	NA	Glycerolipid/MG
TG(18:4/20:5/18:4)	0.004664	0.0458	0.018	0.002	0.240	Glycerolipid/TG
Sucrose Acetate Isobutyrate (SAIB)	0.006477	0.0560	0.008	0.387	0.276	O-glycosyl
PI(22:1/16:0)	0.008388	0.0611	0.017	0.030	0.957	Phospholipid/PI
TG(22:4/20:2n6/22:5)	0.008509	0.0611	0.097	0.303	0.796	Glycerolipid/TG
PG(22:2/20:0)	0.008535	0.0611	0.132	0.144	0.767	Phospholipid/PG
DG(16:1/16:1)	0.009819	0.0631	0.101	0.491	NA	Glycerolipid/DG
DG(22:2/14:1)	0.013877	0.0795	0.014	0.399	0.369	Glycerolipid/DG
2-ethenyl-2,4b,8,8-tetramethyl-tetradecahydrophenanthrene-3,5,10a-triol	0.015343	0.0817	0.100	0.464	0.566	Natural Product
Fargesin	0.018186	0.0895	0.276	0.534	0.155	Psoralen, from Fruits
Ascorbyl linoleate	0.021860	0.1026	0.160	0.358	0.940	Vitamin C metabolite
TG(22:4/15:0/22:4)	0.025945	0.1075	0.255	0.211	0.710	Glycerolipid/TG
TG(22:4/15:0/22:4)	0.028741	0.1146	0.248	0.250	0.824	Glycerolipid/TG
DG(18:4/15:0)	0.030030	0.1154	0.136	0.717	0.462	Glycerolipid/DG
PI(P-16:0/18:3)	0.031557	0.1198	0.131	0.044	0.587	Phospholipid/PI
Ascorbyl linoleate	0.032964	0.1222	0.514	0.164	0.309	Vitamin C metabolite
Ivermectin B1b	0.033354	0.1222	0.110	0.049	0.676	Antiparasitic drug
TG(20:4/15:0/22:5)	0.035029	0.1268	0.912	0.046	0.062	Glycerolipid/TG
Glycidyl oleate	0.037508	0.1293	0.174	0.436	0.891	Carboxylic ester
PE(15:0/20:0)	0.038016	0.1293	0.162	0.575	NA	Phospholipid/PE
PC(14:0/20:5)	0.038145	0.1293	0.147	0.892	0.359	Phospholipid/PC
Ascorbyl linoleate	0.040104	0.1316	0.217	0.333	0.997	Vitamin C metabolite
PC(15:0/18:4)	0.041993	0.1364	0.245	0.294	0.926	Phospholipid/PC
MGDG(18:2/18:3)	0.043317	0.1392	0.398	0.220	0.553	Glycerolipid/MG
Goyaglycoside g	0.043762	0.1392	0.268	0.269	0.860	Cucurbitacin glycosides
Ramipril	0.045603	0.1408	0.991	0.253	NA	Drug
PA(P-16:0/18:4)	0.047472	0.1438	0.518	0.211	0.436	Phospholipid/PA
all-trans-retinyl oleate	0.048268	0.1443	0.388	0.466	0.052	Prenol lipid
PC(14:1/20:5)	0.050091	0.1448	0.196	0.590	NA	Phospholipid/PC
Archaetidylglycerol-myo-inositol	0.052834	0.1499	0.505	0.662	0.281	Lipid

Following database search, 39 compounds from the total 111 compounds found to be nominally associated with color (*p* < 0.05) (Appendix A) were annotated. A total of 11 annotated compounds were found to differ by color with FDR adjusted *p* < 0.05 (above bold line). Compounds are sorted by *p*-value results following linear regression analysis (*p*-value *). Results (*p*-values ^) from pairwise comparisons of pepper color (green vs. red; green vs. yellow, red vs. yellow) are shown with *p*-values < 0.05 highlighted in gray.

**Table 2 metabolites-11-00241-t002:** Qualities of the bell peppers purchased in Denver, Colorado.

Color	Organic Status	Country of Origin	Season	Condition	Replicates
Green	Non-organic	Mexico	Fall 2018	Raw	3
Green	Organic	The United States	Fall 2018	Raw	4
Red	Organic	Mexico	Fall 2018	Raw	3
Red	Non-organic	Canada	Fall 2018	Raw	3
Yellow	Organic	Unknown	Spring 2019	Raw/cooked	2/3
Red	Organic	Mexico	Spring 2019	Raw/cooked	2/3

## Data Availability

All data are available in the manuscript and in the Appendix A.

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
