# Peer review of "Lipidomics-Based Comparison of Molecular Compositions of Green, Yellow, and Red Bell Peppers"

_metabolites, 2021, doi:10.3390/metabo11040241_

Round 1

Reviewer 1 Report

Authors improved the manuscript in accordance with my previous suggestion. The paper can be accepted in present form. 

Reviewer 2 Report

I have revised the first version of this paper, which was submitted as a Research Article. I have seen that the authors have substantially revised the paper, which now is presented as a methodological article. I think that the authors have done a great job in improving the quality of the manuscript, following my first major revision. Therefore, I think that the paper can be accepted in the present form, following minor revisions.

1) The authors have not answered my questions about not using the ESI neg ionization mode. Please, justify this choice.

This manuscript is a resubmission of an earlier submission. The following is a list of the peer review reports and author responses from that submission.

Round 1

Reviewer 1 Report

The present paper describes an untargeted lipidomic approach using LC-MS to identify molecular compounds that can be related to the color of bell peppers. Over the past decades, targeted and untargeted metabolomic (including lipidomic) approaches have gained a lot of interest to identify key metabolic compounds in various raw food materials and food products. Although the topic is of interest, the manuscript should be revised before publication. The low sample size is also an important limit of this lipidomic study as mentioned by the authors and although I understand that it is not always easy to increase the sample size, I would still recommend to do so if possible.

Introduction:

  • The authors put a lot of focus on the biochemical underpinning of the health benefit of food, while the present paper does not investigate these aspects. Using a metabolomic approach to investigate the impact of the molecular composition of bell peppers on their potential health benefit seems to be more a potential perspective/ application of the present method and the introduction should be reformulated as such.
  • The authors reported that the color of bell peppers is largely related to liposoluble compounds such as carotenoids, as previously described in the literature. It would be important to emphasize in the introduction which knowledge gap they aim at filling with their research and which new information is provided by their research/approach that is currently not described in literature.

Materials and methods:

  • The extraction method is missing some information, such as how much raw material was used, how much standard mix was added to each sample, how much solvent mixtures was added per tube at each step or at which concentration was the extracted lipids reconstituted in methanol.
  • The authors mentioned that the hydrophobic samples were spiked with Avanti Polar Lipid Splash Lipidomix. Does this mean that the standard mix was added after extraction? If yes, how did the authors account for potential loss during extraction?
  • The lipid samples were reconstituted in 100% methanol before analysis. Not all lipids are fully soluble in 100% methanol. Was this taken into account in the study?
  • It is unclear which data were used to qualify/quantify the amount of each metabolite. Was it the peak area or a calculated concentration? How was the internal standard mix and the initial amount of raw sample considered?
  • The statistical analysis section is quite succinct and would require more information. It is for example unclear how the pairwise comparison (section 2.4) was performed.
  • The authors mentioned that log transformed data were used for their statistical analysis, while the visualization with PCA was only performed on mean-centered scaled data. Is there a specific reason why the authors did not choose to use log transformed data for the PCA?

Results:

  • Regarding the PCA analysis, the authors present the PCA plot to visualize the different samples cluster based on the color of the bell pepper. It would be interesting to also provide some information from the loading plot to see how strongly the different metabolite analyzed are associated with each component and how they might contribute to the differences in bell peppers from different colors.
  • To my understanding, the HC was performed using the 315 compounds detected in all pepper samples, from which 106 were identified. It would be interesting to take into account this partial identification in the description of the results from HC, in particular in relation to the similarity or difference between the 3 types of bell peppers.
  • Figure 1B: Was the clustering per bell pepper color put in the HC analysis as a pre-requisite by the authors? Or was it the result of the HC analysis?
  • Lines 171-182: It is unclear why the authors performed these analysis, while they clearly stated earlier in their manuscript that due to the low sample size, they would focus on the difference in color only. It would be of interest to keep these information only if they are put into relation with the color differences, i.e. cooking or origin does not affect the relationship between metabolites and color. The authors in this section should also be careful in their conclusion, i.e. it is impossible from their dataset to differentiate the impact of cultivation (organic versus non-organic) from the one of the country of origin (Mexico versus USA versus Canada) and the concentration in non-organic green pepper should not be describe as higher compared to organic green pepper while the results are not significant and the sample size too small to identify a clear trend.
  • It is relatively surprising that beta-cryptoxanthin was the only carotenoids significantly involved in the differences in color. Could the authors comment on that?
  • Although beta-cryptoxanthin is one of the compound the most strongly associated to the color, the analysis highlighted 10 other compounds which are relatively ignored in the description of the results compared to beta-cryptoxanthin. It would be good to balance more the description of the results

Discussion

  • Regarding metabolites associated with the differences in color of bell pepper, the discussion mainly focus on beta-cryptoxanthin, as a pigment compound and on a few selected metabolites that are known to have health effects, while they were not in the top 11 most significant metabolites described in Table 1. Moreover, while the authors clearly stated that due to the low sample size, the focus will be put on metabolites associated with the different colors of bell peppers, the discussion also included a large part on cooking methods and growing conditions. I would recommend to refocus the discussion on the main outcomes and goals of the research.

Other minor comments

  • Line 115: The mention “often from the same pepper” is a little confusing. Do the authors mean from the exact same fruit? Or from bell pepper within the same quality as described in Table 2 (e.g. two different replicates from organic green pepper)?
  • It is unclear which new information is provided in lines 114-118 compared to the information provided in section 2.1. To avoid repetitions, the authors could consider pooling these information together.
  • Caption Figure 1B: The sentence “Blue lines indicate less relative abundance of a compound compared to peppers in other color groups, while red indicate higher relative abundance” would need to be reformulated. If the color was dependent on the comparison with the other 2 color groups, then it seems not possible to have all three groups with dark red or dark blue colors.
  • Figure 1A: The axis x legend (i.e. “Component 1 (15%)”) is missing on the figure.
  • Lines 138-140: The sentence “of these 111 compounds, 39 compounds were annotated following database searching” is a little repetitive as the preceding sentence already mentioned that our of the 111 compounds, 39 compounds were annotated.
  • Line 440-442: Shouldn’t it be “was initially annotates as alpha-cryptoxanthin although manual review of the data showed that beta-cryptoxanthin is indistinguishable from alpha- cryptoxanthin”
  • Did the authors considered coupling the MS detection/annotation with UV detection for example? UV detection can indeed provide information on the absorption spectra of a compound which can help the annotation.
  • Table 1 (Title and footnotes) and Figure 2 (caption): Double dots appear at the end of these and should be changed to one dot only. 

Reviewer 2 Report

The paper presented by Sutliff et al. is aimed to comparison of molecular composition of three types (green, yellow and red ) bell pappers. Scientits focused attention on identyfing and annotating the  molecular composition of papper on human health and tried to determine dependency between the three types of pappers and affectiveness and activity of their secondary metabolites. The subject is interesting and provided information is readable both for scientists and other readers.
There are numerous papers aimed to pappers and identification of their secondary metabolites nevertheless the scientists focused attention on lipidomics, being a sub-field of metabolomics, what made the paper more interesting. 
In my opinion, the paper is well written and studies results are clear. 
Conclusions should be more detailed. Please improve this part of manuscript. 

Reviewer 3 Report

The manuscript entitled: "Lipidomics-based Comparison of Molecular Compositions of Green, Yellow and Red Bell Peppers" by Sutliff et al., describes a very interesting work based on the analysis of pigmented pepper samples and using a lipidomics-based approach to profile hydrophobic compounds. I appreciated a lot the lipidomics-based workflow, when considering both the ID approach (i.e., strong database searching for annotation step) and the LC-QtoF-MS workflow. Overall, the manuscript can be considered for a potential publication following a major revision step.

1) The authors analysed samples from different farming systems (i.e., organic vs conventional ones). We know that the farming system is able to deeply affect the phytochemical composition of a plant-food. Therefore, I suggest to better describe this point in the manuscript.

2) Authors are invited to better describe the extraction step. I don't understand why they decided to discard the hydrophylic fraction following the modified MTBE extraction method. Also, how many mg of lyophilized material was used for the extraction? Please, detail the protocol.

3) Why ammonium formate was not added to the mobile phases? 

4) MS analysis in both ESI+ and ESI- or polarity switching?

5) Authors should highlight the confidence in level of annotation. They used a level 2 (putatively annotation) exploiting high-res MS data.

6) Just a curiosity: why authors did not used Quality Control samples (acquired in tandem MSMS or autoMSMS) to increase the level 2 of confidence for the MS-only data? 

7) I'm an user of Agilent QTOF and the different softwares (Profinder, MPP, Lipid annotator, etc). Please, can you describe more the Profinder parameters and the normalization workflow of mass features in MPP?